# The BowTie as a Digital Twin: How a BowTie Looks Different from a Data Perspective

Paul Singh [1,*], Coen van Gulijk [1] and Neil Sunderland [2]

1   School of Applied Sciences, University of Huddersfield, Huddersfield HD1 3DH, UK; c.vangulijk@hud.ac.uk
2   Syngenta Huddersfield Manufacturing Centre, Huddersfield HD2 1GX, UK; neil.sunderland@syngenta.com
*   Correspondence: p.singh@hud.ac.uk

**Abstract:** This work follows from a research project for safety management system re-engineering that turned a safety BowTie into a digital twin. A digital twin is a model embedded in software that mirrors a specific aspect of a real system; the aspect in this case is the risk space associated with a process. The well-known BowTie is the model that turns out to be singularly well suited as a digital twin from the risk perspective as it maps out the risk space together with real-life controls. However, for a BowTie to be a high-fidelity digital twin of a real system, the rules and processes for designing and operating a BowTie are changed.

**Keywords:** process; safety; monitoring

## 1. Introduction

There is no common definition for the term big data; however, it can be described as the process of dealing with actual, differing, large volumes generated at a fast pace that is of value to an organization [1,2]. Chemical process plants produce petabytes of sensor data that are accessible but with some challenges which are largely due to the lack of cost-effective data processing solutions. The big data challenges include:

1.   Storage;
2.   Management;
3.   Analyses.

Chemical processing plants operate with distributed control systems (DCS) or with programmable logic controllers (PLC). This control system is a set of equipment, through the use of control loops, that regulates devices or systems. Control loops consist of components that measure (with a measuring element—thermocouple, venturi tube, differential pressure device, etc.) the controlled variable (the measurement that is meant to be controlled) and transmit that measurement to a controller (usually an algorithm), which then instructs the final control element (e.g., a valve) with the desired response. Data generated by the process control systems are stored in a variety of storage systems. Some examples of the organizations that provide solutions for data lakes or process historians are:

1.   AVEVA PI system [3];
2.   Honeywell Experion systems [4];
3.   Yokogawa data historian [5];
4.   dataPARC [6];
5.   Automsoft [7].

The chemical industries are served with various commercial software solutions that are optimized for working with industrial scale data in data lakes [8]. Extracting the right data from data lakes requires (a) knowledge of the process system, which is generally supplied by chemical engineers; (b) knowledge of the data storage systems, which is usually supplied by data analysts; and (c) a scalable processing code that facilitates consistent and repeatable access. The latter is often a practical problem; scientists and programmers

can create bespoke solutions; however, that is usually not an optimal solution to serve a corporation controlling dozens of chemical plants on multiple, potentially global, sites. Earlier work shows that SeeQ [9] offers consistent data extraction methods, as well as navigation between the information technology (IT) and operational technology (OT) landscapes; these methods are sufficiently accessible for chemical engineers to program themselves. SeeQ is a commercially available software application which allows analysis of the time series data generated by a process. The application comprises three components: the workbench, organizer, and data lab [10]. The data are not stored in any additional systems; however, the data are fed through the application, and the results from the calculations, via the SeeQ Cortex, are shown in the application itself.

As identified by Busch et al. [11], the interpretation of the data to produce information that leads to action is difficult. Dealing with that difficulty can be fruitful for the organization because, as identified by the World Economic Forum, data are an economic asset [12,13]. From a safety management perspective, another tool is required to present the information that is generated by the data analyses. The time series data generated by the process stored in the data lake can now be used directly and presented using a visual tool, the BowTie, to produce a high-fidelity system. With simple counts and measures, the information can be readily understood by all levels of personnel throughout the organization.

Our earlier paper explains [14] how chemical engineers can overcome traditional software shortcomings; this puts them in a better position to exploit the value locked in data lakes and to recast their traditional views on process safety management. This paper goes beyond that earlier paper by demonstrating how the technology is used to turn a traditional BowTie into a digital twin for process safety.

Section 2 provides the background theory to the elements of a BowTie and a digital twin. Section 3 explains how to create a BowTie digital twin, in theory. Section 4 provides an example of the creation of a BowTie digital twin and presents the results and discussion. The conclusions from the creation of the BowTie digital twin, leading to the recommendations and the path forward, are given in Section 6.

## 2. Literature

### 2.1. BowTies

The individuals responsible for the management of process safety can use BowTie diagrams to model the hazards and top safety events which may occur in their respective processes. The BowTie model showcases the threats which may lead to the top event and, on the left side of the diagram, any preventative barriers that may be in place. From the top event to the right of the diagram, the mitigating barriers which are in place to reduce the realization of the occurrence of the final consequence are showcased. Those final consequences may be identified using "PEAR" criteria (consequences with respect to people, the environment, assets, and organizational reputation) [15]. The major components of a BowTie are:

1. Threats;
2. Preventative barriers;
3. Hazard/Top Event;
4. Mitigating barriers;
5. Final consequences (as per PEAR).

The BowTie model then becomes the linear visualization of the safety management system of the process [16,17]. Data quality has been identified as a main issue for BowTies as the BowTies represent complex environments, most often for rare events [17]. Figure 1 below shows a general representation of a BowTie diagram.

BowTie diagrams can be connected to data and can be used to compute respective probabilities in order to provide an indication of system integrity and reliability [17,18].

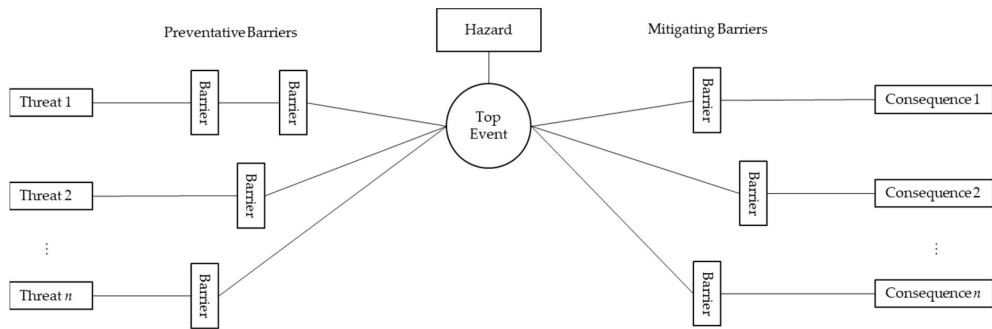

**Figure 1.** Generic BowTie diagram.

To create a BowTie diagram, a working group starts with a particular process and identifies the appropriate hazard (the component that has the potential to cause harm) and the top event under investigation. Depending on the mandate and direction provided by the organization, the consequences can be determined by analyzing the worst-case scenarios with no mitigations or safeguards with respect to the people, environment, assets, and reputation. The threats that may lead to the actualization of the top event can be identified from other safety investigations, such as process risk assessments (PRAs) or hazard and operability (HAZOP) studies, for example. Preventative barriers can be identified by reviewing the measures in place that are capable of preventing the occurrence of the top event. For the mitigation barriers, once those worst-case consequences are identified, the working group can then determine the mitigations that are currently in place. Additional information can then be included, like the degradation factors and controls, such as maintenance and inspections, that exist for those barriers. The team can then review the BowTie diagram and move on to the next hazard and top event. Figure 2 below summarizes the steps required for creating a BowTie diagram for a particular process.

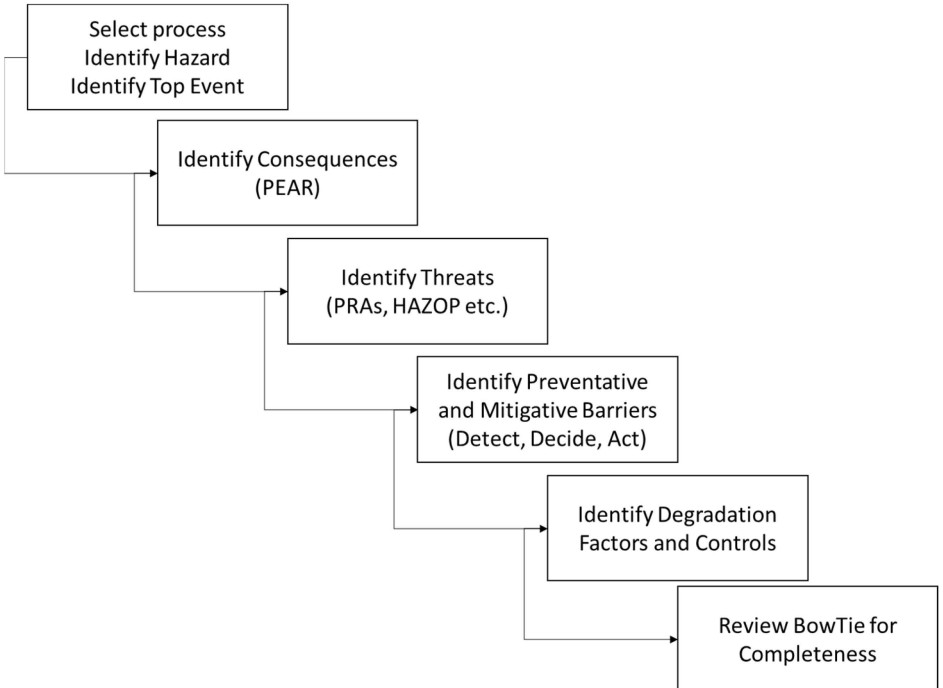

**Figure 2.** Steps for creating a BowTie diagram.

For this paper, the main question with regard to the digital re-engineering of the BowTie was: how can we turn the BowTie into a digital twin?

*2.2. Digital Twin*

Digital twins are commonly referred to as digital systems, which are used to replicate real systems [19]. Everything begins with the study of a real system with a certain functionality. In our case, this is the safety management system of a process. This real system comprises physical components and algorithms which process the data generated by the process. These physical components are the hardware of the system. The software and control algorithms are located in an information process system (IPS), which is typically an operating system. Then, the real system is connected via a network or communication system. The digital twin of the real system is created in a virtual environment to emulate the real system.

Commonly, the credit for the conceptualization of digital twins is given to Dr Michael Grieves with the concept of Product Life cycle Management (PLM) back in 2002 [20,21]. The term digital twin was coined by John Vickers of NASA [20]. Whilst working on digital twins, Grieves and Vickers understood that advances in technologies made systems increasingly more complex.

It is beyond the scope of this paper to discuss the numerous definitions of digital twins which describe different features of digital twins. This definition from Nath suffices for this practical development of an actual digital twin:

> "A digital twin is a synchronized instance of a digital template or model representing an entity in its life cycle and is sufficient to meet the requirements of a set of use cases." [20]

The definition showcases the requirement for a model of an "entity" and shows that this digital twin must satisfy some business need. The entity in the definition does not need to be a physical asset; as defined by the International Organization for Standardization (ISO), an

> "entity is an item that has recognizably distinct existence, e.g., a person, an organization, a device, a subsystem, or a group of such items." [22]

Digital twins became an accepted approach for gathering additional awareness and understanding of operational and maintenance issues [23]. For example, aside from data analytics, physics-based algorithms can be used along with historical data to create simulations and allow predictions of system behavior. The fusion of digital and physical spaces leads to "flexibility and scalability" [23].

To create a digital twin model diagram, a working group starts with an asset and defines the purpose that the digital twin is trying to satisfy within the organization. The asset may need to be broken down into component parts, including the asset and the subsets of the asset. Data elements are then required to establish the data hierarchy between the subsets and sets, which then allows the creation of the digital twin. Once constructed, the working group can now align the data flows from the asset to the digital twin. This will then allow validation of the digital twin, which could lead to a multitude of benefits for the organization (cost savings, etc., as per the goal or purpose of the digital twin). Figure 3 below summarizes the steps required to create a digital twin model for an asset.

A digital twin of any process has the potential to lead to the continuous improvement of a process. When reviewing the BowTie and understanding the premise of a digital twin, the subsequent question to be asked is related to the next step that needs to be taken to upgrade a BowTie to a BowTie digital twin.

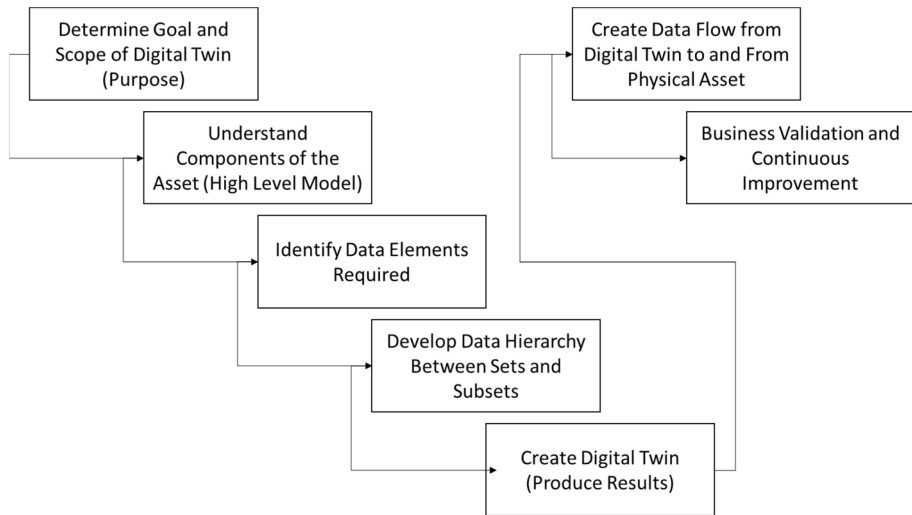

**Figure 3.** Steps for creating a digital twin model.

## 3. BowTie Digital Twin (BTDT)

### 3.1. Theoretical Background for a BowTie as a Digital Twin

From the definition of a digital twin, the BowTie would visually represent the safety management system that is already installed in the plant to deal with the particular top event which arises from a hazard. This *is* the model for the physical safety management system. Alternatively, safety instrumented systems (SISs) are sets of hardware and software designed to return processes back to a safe state by shutting down processes if hazardous conditions are measured [24]. The SIS status can be displayed on a DCS to alert operators if a response is required [25]. A safety integrity level (SIL) rating indicates the SIS reliability through calculations of probability of failure upon demand [26]. These systems must operate independently to ensure their reliability [25]. As per IEC61511, the overall safety of the process depends upon the setup of the basic process control system (BPCS) as well as the correct functioning of the SISs [25,27]. The BowTie visually represents all barriers, not just the SISs; however, the SISs can also be barriers in a BowTie diagram.

For many organizations, not all barriers have data tags or associated auto-generated data. For example, a barrier which requires human intervention, or other barriers without sensors collecting data, such as near-miss or incident reports, may not be stored in a system where the data can be easily retrieved, such as an individual's computer system using a manually updated spreadsheet [28]. These barriers need to be identified and the data associated with them need to be located and cleansed so that they are stored in a central system and then aligned (i.e., time-stamped) to allow the analysis to occur.

Typically, there are multiple instances of barriers identified in the BowTie which represent multiple assets. Theoretically, the barriers follow the 'detect/decide/act' definition of a barrier [15]. Most of the detectors are sensors or measurement devices used to detect temperature, pressure, level, flow, or any other fluid property. Decision tools are normally algorithms used to instruct a final piece of equipment (i.e., a valve) to perform an action. As per the definition of a digital twin entity, these barriers can represent assets of the same type. Finally, according to the definition of a digital twin, the BTDT addresses a specific business challenge related to ensuring that the process, plant, or equipment operates within a safe envelope. Expanding the definition of an entity further, the barriers themselves do not have to be simply physical assets. The barriers can also be people, policies, and procedures.

Figure 4 below shows how the operational processes flow for the continuous operation of a BowTie as a digital twin. Now the fact that BowTies can be digital twins has been established, how does one go about creating a BowTie digital twin?

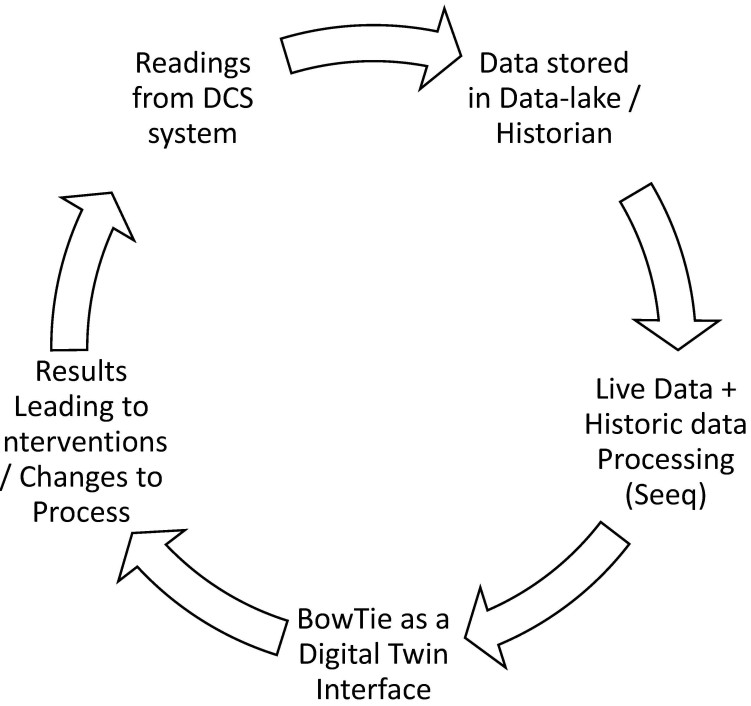

**Figure 4.** BowTie digital twin operations.

*3.2. Creating a BowTie Digital Twin*

    To create a BTDT, the members of a working group start a particular process just as if they were beginning with the creation of the BowTie and then continue with that process until the BowTie has been validated by the working group. In essence, this means that the processes in Figures 2 and 3 merge into the process drawn in Figure 5. With an acceptable starting BowTie, the data elements are then identified and located within the data lake. This is where the specialized software (SeeQ R50) is used to allow the working group to traverse through the IT/OT landscape. Once the elements and data hierarchy are set, the BowTie can become the digital twin of a particular process. A dashboard can also be created to showcase any information that is produced by the BTDT to allow operators or management to make the decisions that then alter the process. This represents the data flow from the digital twin to the asset. The data generated by the process then feed the digital twin, which, as mentioned earlier, could lead to a multitude of benefits.

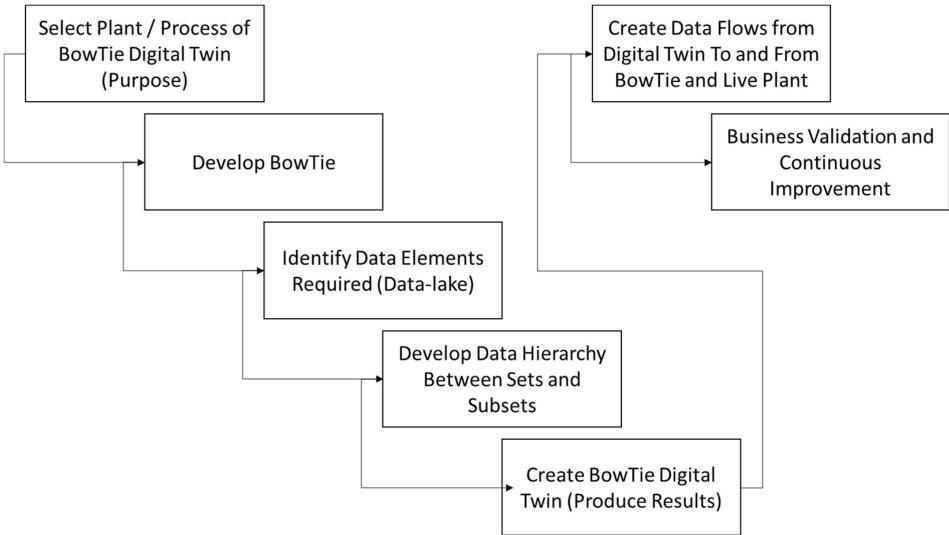

**Figure 5.** Steps to create a BTDT.

### 3.3. BTDT Project Phases

The method capitalizes on prior work that developed a functional solution architecture to connect BowTies to a data management system that created a simultaneous rapid overview and inspection of all the relevant barriers [14]. The solution architecture can be based on software solutions such as the SeeQ software solution [10]. The BowTie is shown with links to management tasks. The BowTie represents a flow diagram that connects hazards, barriers, the potential accident, and the consequences in a quick graphical overview. The barriers are based on safety business processes and data streams that can be used to populate the BowTie.

To create a BTDT, a project could contain a minimum of five steps to achieve the aims, as shown in Table 1 below:

**Table 1.** Project steps.

| Step | Problem | Task | Results |
|------|---------|------|---------|
| 1 | What do you want to control? | Selection of a barrier | High-fidelity BowTie<br>Identify data sources<br>Barrier(s) selected |
| 2 | Where do you monitor? | Detect–decide–act modelling | SeeQ monitoring systems<br>Initial tests with data source |
| 3 | How to aggregate? | Develop key performance indicator (KPI) for the selected barrier | Monitoring<br>Systematic rational for KPIs<br>KPI for barrier(s) |
| 4 | When do you visualize? | Collate user requirements | Design of communication (software) requirements |
| 5 | Where do we go from here? | Capture lessons learned | Work completed<br>Benefit projections<br>Roadmap for scale-up<br>Cost projections |

The foundation has been set. A BTDT is possible.

## 4. Example BTDT

### 4.1. Process

This work follows on from the case study at Syngenta's Huddersfield Manufacturing Centre (HMC) [29], located in Huddersfield, West Yorkshire, UK, as described by Singh et al. [14]. The reaction involves two reactants as well as two catalysts. As mentioned by Singh et al. [14], the process reaction rate is controlled with the addition of the second catalyst. The simplified process flow diagram is shown in Figure 6 below:

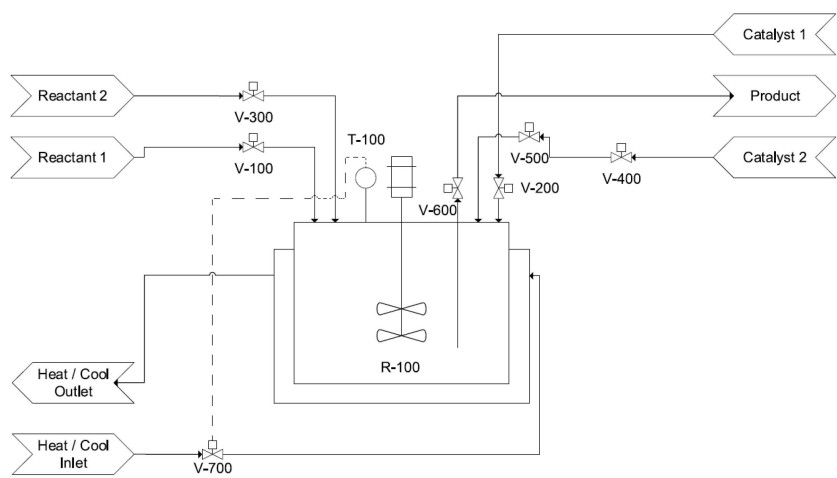

**Figure 6.** Syngenta HMC simplified process flow diagram (source: Singh et al. [14]).

Upon review of the process, the next development is the review of the BowTie for the process.

### 4.2. Syngenta HMC BowTie Redevelopment

The original BowTie had to be reevaluated to create the new BTDT by using the flowchart, as presented by Singh et al. [14], in conjunction with Figure 6 above. The original BowTie showcasing the threats to the top event and subsequent consequences is reproduced Figure 7 below.

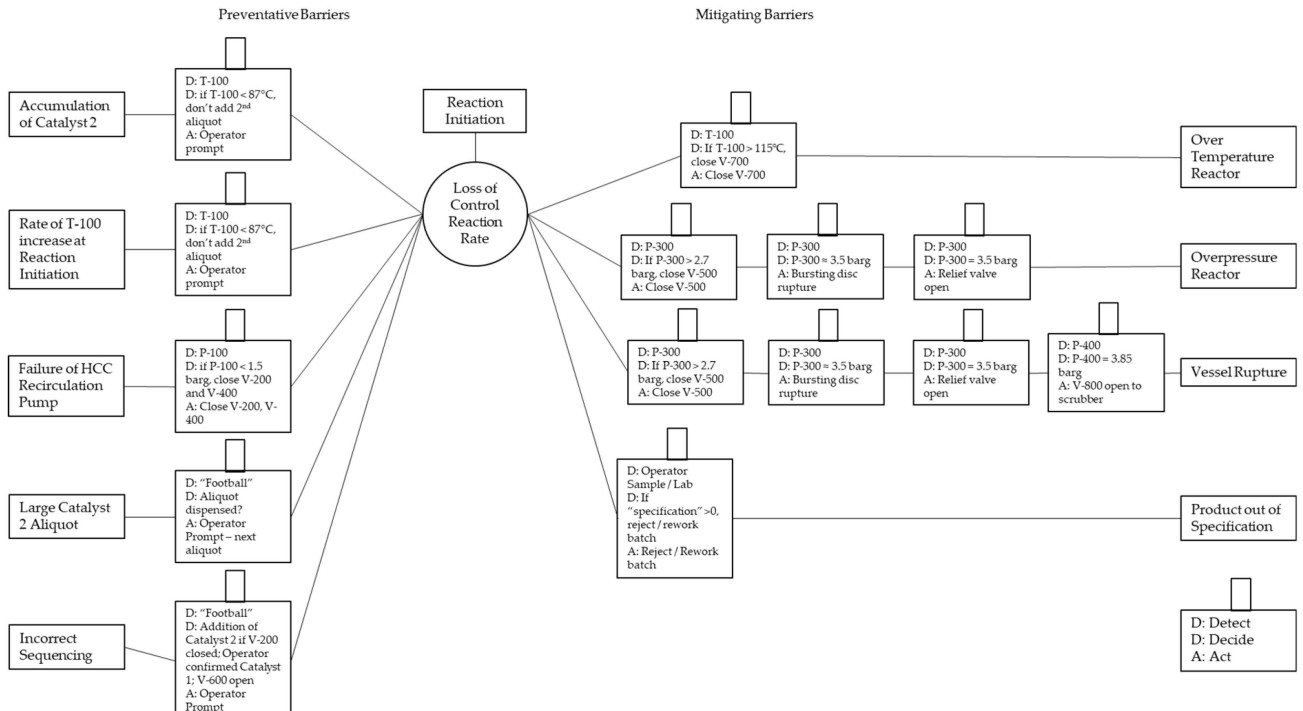

**Figure 7.** Traditional Syngenta HMC stage 3 reaction BowTie (adapted from Singh et al. [14]).

A point to note is that not all the equipment identified in the BowTie in Figure 7 is displayed in the simplified process flow diagram shown in Figure 6. The BowTie review process begins with the preventative barriers. The first threat shown in the BowTie describes the scenario in which catalyst 2 is charged and has not reacted and the next aliquot has been added; this may lead to a sudden rise in the reactor temperature, which could ultimately lead to an overpressure event and a subsequent vessel rupture. The detection for the first threat is not conducted with respect to the reactor temperature, as initially conceived. Instead, since the threat occurs with respect to the accumulation of catalyst 2, the detection should be conducted with respect to the inactivity of catalyst 2 during the first aliquot addition. This detection can be conducted with the observation of whether the temperature of the reactor decreased prior to the addition of the second aliquot. For the decision and action components of the barrier, if there was an initial charge and no reduction in temperature was observed, then the operator should not proceed with the second aliquot.

The second threat, which involves the rate at which the temperature of the reactor increases at reaction initiation, showcases the same barrier as the first threat. The previous analysis completed by Singh et al. [30] shows that this barrier is not a barrier; alternatively, however, it can be monitored and displayed as a separate KPI for the process itself. Therefore, the threat and barrier were removed from this BowTie.

The failure of the heat–cool–chill (HCC) pump could lead to a loss of control of the reaction rate as the operators would not have control over the process temperature, which could lead to an overtemperature event. The barrier associated with this threat follows the simple barrier described by the Center for Chemical Process Safety (CCPS) [16]. The failure

of the HCC pump can be detected by measuring the pressure at the outlet of the pump, and if the pressure falls below the prescribed threshold of 1.5 barg, then the routing valves for catalyst 1 and catalyst 2 are closed.

Threat 4, listed in the original BowTie, refers to the threat of a large addition of the second catalyst. The large amount of catalyst 2 could also lead to a loss of control of the reaction rate, culminating in an overtemperature event. The barrier revolves around the size of the aliquot, which is referred to as the "football" by the organization. The decision and subsequent action originally centered on whether the aliquot was dispensed and led to an operator action. Upon further review, the organization must determine the size threshold of the aliquot to define the term "large", which could lead to a loss of control of the reaction rate. Likewise, if there was a large aliquot dispensed, then the subsequent catalyst charge should not be permitted. If this aliquot addition was the initial addition of catalyst 2 to the reaction, then the action would be identical to threat 1 with regard to the accumulation of catalyst 2.

Threat 5 is related to the incorrect order of the catalyst addition to the process. If catalyst 2 were to be added to the process prior to the addition of catalyst 1, then this situation could lead to the loss of control of the reaction rate, leading to an overpressure event and subsequent vessel rupture. Initially, the BowTie makes a reference to the detection of the "football". The initial detection should be related to the presence of the addition of catalyst 1. If the catalyst 1 charge chute is open and the operator has not yet confirmed the addition of catalyst 1, then the process should not allow the addition of catalyst 2.

Going forward, the same analysis and review to allow simplification can be performed with respect to the mitigation barriers. The review considers the consequences that impact the people, environment, assets, and reputation (PEAR) of the organization. This type of review, the PEAR review, is similar to the reviews conducted when analyzing the hazards and risks associated with processes. From a safety management perspective, the barriers are placed in a chronological order with respect to severity or impact.

The processes are designed with inherent safety in mind so that the environmental, asset, or personnel risks are minimized as much as possible initially; one of the first intentions of organizations, from an operational perspective, is to ensure that their reputation is intact and that the commitment to customers is upheld. When chronologically describing a process, one of the first mitigation barriers to come into action is the prevention of reputational damage to the organization. This is typically where organizations take samples from their processes for quality assurance and quality control purposes. In this instance and for this process, the original assumption was that samples were taken from the process to ensure that the product from the stage 3 reaction was within the specification. However, this does not hold true. The stage 3 reaction is no longer sampled and tested; however, since the product of the stage 3 reaction is an intermediate product for the overall process, if the product is not within the specification, it may be re-worked, which just leads to extra processing time. The impact on the company reputation would be a delay in production, leading to subsequent delay in delivery to the customer. Additional mitigation to prevent the consequences of a loss of reputation involves the fact that batches of the final product are produced in another part of the plant; therefore, any delays may be eased by the storage levels of the final product. With respect to this particular barrier for the stage 3 reaction, the analysis would not involve taking a sample to see if the product of the reaction was within the specification; it would be the time required for processing. The barrier would be as follows:

1. Detect—Time required for stage 3 reaction;
2. Decide—Time required is within an acceptable threshold (the organization would need to determine the acceptable threshold for the stage 3 reaction);
3. Action—If the time required for the stage 3 reaction exceeded the threshold, mitigation would involve ensuring that the final product storage levels were adequate to satisfy customer demand.

Progressing with the BowTie, the consequence lines leading to an overtemperature event of the reactor ultimately lead to an overpressure event, which could lead to vessel rupture. Likewise, an overpressure event would also lead to vessel rupture. Of the mitigation barriers identified in the original BowTie for this process, the barrier with the final action around the bursting disc is not a true barrier. As explained by experienced operators of this process, the intention of the bursting disc is to act as a physical protection device for the relief valve as the reactor material is corrosive to the relief valve. Without placing the process under additional risk, the bursting disc is rated to a pressure rating just below that of the relief valve so as not to prevent the relief valve from performing its intended task. Therefore, the barrier with the bursting disc identified as the active element is removed.

Returning to the overtemperature event, if the temperature threshold of 115 °C is exceeded, the pressure does not go above the 3.5 barg needed to trigger the relief valve. However, the initial barrier to prevent further addition of catalyst 2 would have already been triggered. From a review perspective, the safety management system is in place to prevent an overpressure event leading to vessel rupture with an early preventative measure that does not allow the further addition of catalyst 2 to the process. With catalyst 2 involved in the reaction, the temperature of the reactor may still rise to a point that exceeds 115 °C.

All barriers are in place to prevent the ultimate final consequence of a vessel rupture, which could lead to personnel injuries or casualties. The consequence line can be summarized as a single line towards a final end.

With all the points above, the newly updated BowTie can be summarized in Figure 8, as follows:

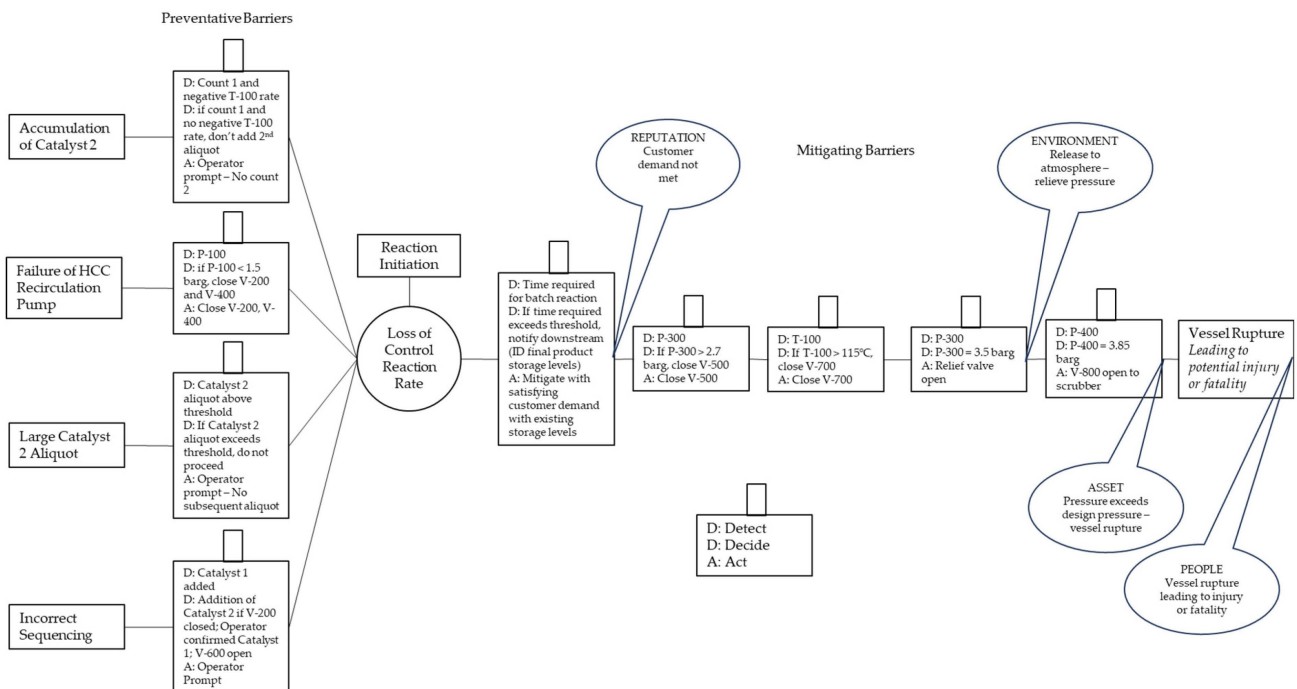

**Figure 8.** A BowTie looks different from a data perspective: Syngenta HMC stage 3 reaction BowTie digital twin (BTDT).

As the BowTie has been revamped, all of the required requisite models can be developed.

### 4.3. Model Development with Data Elements and Hierarchy

#### 4.3.1. Preventative Barrier 1

By reviewing the updated BowTie in Figure 8 in the previous section, it can be seen that model development begins with the first threat: the accumulation of catalyst 2, as listed

from top to bottom in the left-hand side of the diagram. The barrier to deal with this threat is an active and human barrier, as defined by CCPS [16]. The primary data tag required is the reactor temperature. The logic associated with this barrier is that if a temperature decrease is not witnessed during the first aliquot addition, the operator should not add the second aliquot.

The model development required the addition of the temperature measurement data, T-100. The data were smoothed out using an agile filter based upon the Loess method [31] to reduce any potential erroneous results; Further data manipulation was conducted using the formula, which requires an output sample rate and a timing window length, as shown below:

$$\$T100.agileFilter(5\text{ s},\ 3\text{ min}) \tag{1}$$

The derivative function is then used on the resultant smoothed temperature reading:

$$\$T100.derivative() \tag{2}$$

If a derivative function had been used on the original data, many spikes would have been noticed due to the nature of the function built into the software. As described by Singh et al. [30], the derivative function is calculated using the current and previous sample points. If there was a significant shift between two simultaneous measurements, then a spike would be observed in the subsequent derivative signal. Upon smoothing, those spikes are eliminated to provide a better indication of the rate of change of the temperature of the reaction for the process. A condition is then created from the resultant derivative signal to identify any negative temperature trends. The count signal needs to be changed to a condition that allows simultaneous analysis of the temperature rate and the aliquot count. The following function allows conversion of the signal to a condition and keeps the count between the requisite aliquot values:

$$\$Catalyst2Count.toCondition('Count'.keep('Count',\ isbetween\ (1,\ 31))) \tag{3}$$

The barrier is checked by viewing the rate of change in the temperature readings during the first aliquot and by checking whether a second aliquot exists. A composite condition is created to perform that check. As described by the CCPS [16], this type of barrier prevents the threat from propagating into the central event.

### 4.3.2. Preventative Barrier 2

The active barrier to deal with the threat of a low-pressure event in the HCC unit prevents a possible loss of control of the reaction rate. The primary data tag required is the outlet pressure from the HCC pump, P-100. The logic associated with this barrier is that if the outlet pressure drops below 1.5 barg, then the catalyst routing valves are closed to prevent any further addition to the reactor. The addition of a catalyst may prompt reaction initiation, and a hazardous situation may develop if there was no adequate cooling supplied to the reactor during the stage 3 reaction.

The model development required the addition of the pressure measurement data, P-100. The conditions were then created to identify any instances where low pressure was observed during the stage 3 reaction. The barrier was checked by counting the number of times the low-pressure events occurred and whether the catalyst routing valves were closed during those instances. If no results were present during the stage 3 reaction, the analysis could be extended to include all data points to capture any results from the maintenance barrier testing.

### 4.3.3. Preventative Barrier 3

The barrier to deal with the threat of a large catalyst 2 aliquot addition was another active and human barrier. The primary data tag required was the mass of the catalyst 2 aliquot that was dispensed. The logic associated with this barrier is that if the size of the

catalyst 2 aliquot dispensed is above the allowable threshold, the operator should not add a subsequent aliquot.

The model development required the addition of the mass of the aliquot, M-100. To ensure that the data lied only during the stage 3 reaction, the R100PhaseName data were required, as well as the Catalyst2Count tag. As shown in Equation 3 above, the count signal was converted to a signal in order to then be able to determine the maximum value of each aliquot addition, as the M-100 signal shows the mass of the contents of the vessel feeding the aliquot to the reactor. The following formula allows the determination of the maximum value of each aliquot dispensed during the stage 3 reaction:

$$\begin{aligned}(\$Catalyst2Count.removeLongerThan(60\,\text{h})).\\setProperty('MaxValue',\$M100,\ maxValue())\end{aligned} \tag{4}$$

The "*removeLongerThan*" code is required as it is a constraint within the software, and a maximum duration is required to use the formula that allows one to determine the maximum value of a signal during a condition. The resultant maximum value is then converted to a signal:

$$\$Catalyst2MaxValue.toSignal('Max\ Value') \tag{5}$$

The signal profile can be used to create a reference profile to compare with future aliquot additions. To create the reference profile, the signal must be merged to limit the gaps between the data points.

$$\$Catalyst2MaxValue.merge(1\,\text{s}) \tag{6}$$

The size of the aliquot is required to determine whether the aliquot size was exceeded. There is a requirement to have logic built in where the operator cannot proceed with the next aliquot, which can be conducted using the system itself. For example, if the size of the aliquot exceeds the threshold, the catalyst 2 routing valves are closed. Currently, this barrier does not exist. If the barrier was surrounding the initial aliquot size (count 1), then the same logic as that shown in first barrier could apply (by treating the large aliquot as an accumulation of catalyst 2).

### 4.3.4. Preventative Barrier 4

If catalyst 2 is charged before catalyst 1, this could lead to a loss of control of the reaction rate and the subsequent overpressure of the reactor. Before beginning the batch, the regular position of the valves is closed; the charging vessel (i.e., the football) contains catalyst 2, and the temperature of the reactor is at an ambient temperature. It is also worthwhile noting that the operators are validated to run the process; therefore, the operators understand that catalyst 2 cannot be charged before catalyst 1.

The main tag required for this model is the string data tag, R100OpMSG, which presents the main reactor operator prompt messages. The other data tags required are the catalyst routing valves' positions. A condition is created to identify when catalyst 1 is charged during the reaction and when the operator confirms the addition of catalyst 1. Another condition is required to identify when the catalyst 1 charge chute is also open. Once again, the operator then confirms the next step as the system waits for the depression of the pushbutton. To determine the functionality of the active plus human barrier, one can confirm that the catalyst 1 charge chute valve is open during the operator confirmation of the charge. This can be conducted using the following formula:

$$\$Catalyst1PushButton.intersect(\$V200 = 0) \tag{7}$$

Table 2 below summarizes the components of the four preventative barriers.

**Table 2.** Preventative barrier summary.

| | Threat | Primary Data | Logic and Action | Barrier Test | Barrier Type (CCPS) | Data | Action |
|---|---|---|---|---|---|---|---|
| 1 | Accumulation of Catalyst 2 | T-100 | If T-100 does not decrease, do not add aliquot 2 | Determine if T-100 temperature decrease and second aliquot added | Active + Human | All tags | None |
| 2 | Low HCC Pump Pressure | P-100 | If P-100 ≤ 1.5 barg, close catalyst 2 routing valves | Determine if P-100 ≤ 1.5 barg and routing valves closed | Active | All tags | None |
| 3 | Large Aliquot Catalyst 2 | Catalyst 2 mass | If catalyst 2 mass greater than threshold, do not add subsequent aliquot | Determine if mass of aliquot within threshold and subsequent aliquot added | Active + Human | All tags | Mass threshold |
| 4 | Incorrect Sequencing | Catalyst 1 | If catalyst 1 charged, then allow catalyst 2 | Determine operator confirmation of catalyst 1 addition | Active + Human | All tags | None |

Going forward, the same methodology is applied to the mitigation barriers.

### 4.3.5. Mitigation Barrier 1

If the amount of time required for the stage 3 reaction changes over time, it may lead to a reputational risk for the organization as delays at this stage of product production may have knock-on effects on downstream processes and may ultimately result in the inability to satisfy customer demand. For this barrier, the stage 3 reaction initiation is identified with the tag, R100PhaseName. As mentioned for the previous barriers described above, this tag produces a string result with several values. The value of the "Stage 3 Reaction" shows when the reaction was initiated, the progression of the reaction, and, finally, when the reaction was completed. The signal can be used to show when the batch started and when the operation has been completed. To perform this analysis, the R100PhaseName signal is given to a condition and that condition is renamed "Reaction" by using the following formula:

$$\$R100PhaseName.toCondition('Reaction') \tag{8}$$

The new condition can now be converted to a new "Duration" signal which allows the determination of the length of the stage 3 reaction:

$$\$PhaseNameCondition.toSignal('Duration', DurationKey()) \tag{9}$$

To determine the health of this reputational barrier, a value search is required to identify when the duration of the stage 3 reaction exceeds an acceptable threshold. This threshold is to be determined by the organization.

### 4.3.6. Mitigation Barrier 2

The active barrier to deal with the mitigation of a high-temperature event is to prevent a vessel rupture event which may lead to a potential casualty during the process. The primary data tag required is the R-100 reactor temperature, T-100. The logic associated with this barrier is that if the temperature exceeded the threshold of 115 °C, then the steam inlet valve, V-700, would close to prevent additional heat being provided to the reactor.

The model development required the addition of the temperature measurement data, T-100. The conditions were then created to identify any instances where high temperature was observed during the stage 3 reaction. The barrier was checked by counting the number of times the high-temperature events occurred and whether the steam inlet valve was closed during those instances. As with the other barriers, if no results are present during the stage 3 reaction, the analysis can be extended to include all data points to capture any results from the maintenance barrier testing.

### 4.3.7. Mitigation Barrier 3

The active barrier to deal with the mitigation of a high-pressure event is to prevent a vessel rupture event which may lead to a potential casualty during the process. The primary data tag required is the R-100 reactor pressure, P-300. The logic associated with this barrier is that if the reactor pressure exceeds the threshold of 2.7 barg, then the catalyst 2 routing valve, V-500, would close to prevent any additional catalyst 2 in the reactor. This barrier was the precursor to the release of any material to the environment.

The model development required the addition of the pressure measurement data, P-300. The conditions were then created to identify any instances where high pressure was observed during the stage 3 reaction. The barrier was checked by counting the number of times the high-pressure events occurred and whether the catalyst 2 routing valve was closed during those instances. Once again, if no results are present during the stage 3 reaction, the analysis can be extended to include all data points to capture any results from the maintenance barrier testing.

### 4.3.8. Mitigation Barrier 4

The active barrier to deal with the mitigation of a high-pressure event is to prevent a vessel rupture event which may lead to a potential casualty during the process. Once again, as with the barrier mentioned above (mitigation barrier 3), the primary data tag required is the R-100 reactor pressure, P-300. The logic associated with this barrier is that if the reactor pressure exceeded the threshold of 3.5 barg, then the relief valve would open to relieve the reactor pressure by venting into the atmosphere. This is the last resort prior to vessel rupture or to the reduction in the potential loss of assets.

The model development required the addition of the pressure measurement data, P-300. The conditions were then created to identify any instances where high pressure was observed during the stage 3 reaction. The barrier was checked by counting the number of times the high-pressure events occurred and whether the relief valve was opened during those instances. To determine whether the relief valve had opened, a secondary pressure trigger switch (P_HI) was used to show when that trigger was activated in order to then open the relief valve. Once again, if no results are present during the stage 3 reaction, the analysis can be extended to include all data points to capture any results from the maintenance barrier testing.

### 4.3.9. Mitigation Barrier 5

The final active barrier is the last resort prior to vessel rupture; it allows the safe evacuation of all personnel in the surrounding area. The primary data tag required is the R-100 reactor pressure, P-300. The logic associated with this barrier is that if the reactor pressure exceeded the threshold of 3.85 barg, then the vent valve to the scrubber would open to provide a final attempt to relieve the reactor pressure and to allow the safe evacuation of any personnel.

The model development required the addition of the pressure measurement data, P-300. The conditions were then created to identify any instances where high pressure was observed during the stage 3 reaction. The barrier was checked by counting the number of times the extreme-pressure event occurred and whether the vent valve was opened during those instances. Once again, if no results are present during the stage 3 reaction, the analysis can be extended to include all the data points to capture any results from the maintenance barrier testing.

Table 3 below summarizes the five mitigation barriers for the stage 3 reaction.

**Table 3.** Mitigation barrier summary.

| | Mitigation | Primary Data | Logic and Action | Barrier Test | Barrier Type (CCPS) | Data Availability | Action Required |
|---|---|---|---|---|---|---|---|
| 1 | Reputation | PhaseName | If stage 3 reaction duration exceeds threshold, operator to notify downstream operations | Determine when stage 3 reaction is within threshold and no operator prompt | Active + Human | Operator notification | Operator notification data stream |
| 2 | High Temperature | T-100 | If T-100 ≥ 115 °C, close V-700 | Determine when T > 115 °C and V-700 closed | Active | All tags | None |
| 3 | High Pressure | P-300 | If P-300 ≥ 2.7 barg, close V-500 | Determine when $p$ > 2.7 barg and V-500 closed | Active | All tags | None |
| 4 | Environmental Release | P-300 | If P-300 ≥ 3.5 barg, open Relief Valve | Determine when $p$ > 3.5 barg and P_HI trigger relief valve | Active | All tags | None |
| 5 | Assets and People | P-300 | If P-300 ≥ 3.85 barg, open V-800 | Determine when $p$ > 3.85 barg and V-800 open | Active | All tags | None |

*4.4. BTDT Data Flows and Results from BTDT*

For the second mitigation barrier, where the detection is performed by T-100, the logic associated with the barrier is that if the temperature of the reactor were to exceed 115 °C, then the low-pressure steam inlet valve should close. Figure 9 below shows the data of the valve position, the temperature of the reactor, and the stage of the reaction, as described in the methodology above.

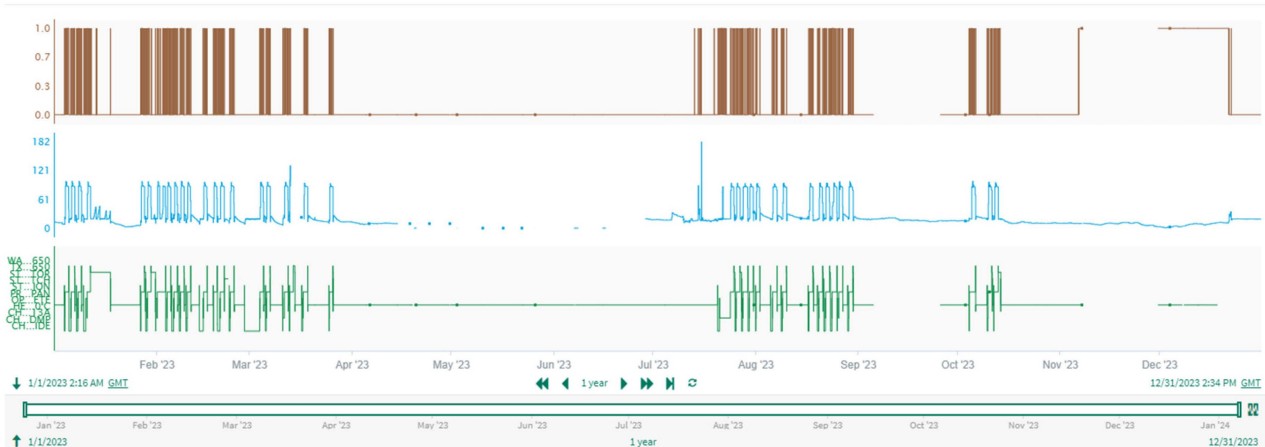

**Figure 9.** Tag data for 2023 calendar year for mitigation barrier 2.

From the data presented, there are two occasions, noted by the red circles, where the barrier was required during the 2023 calendar year, as shown in Figure 10 below:

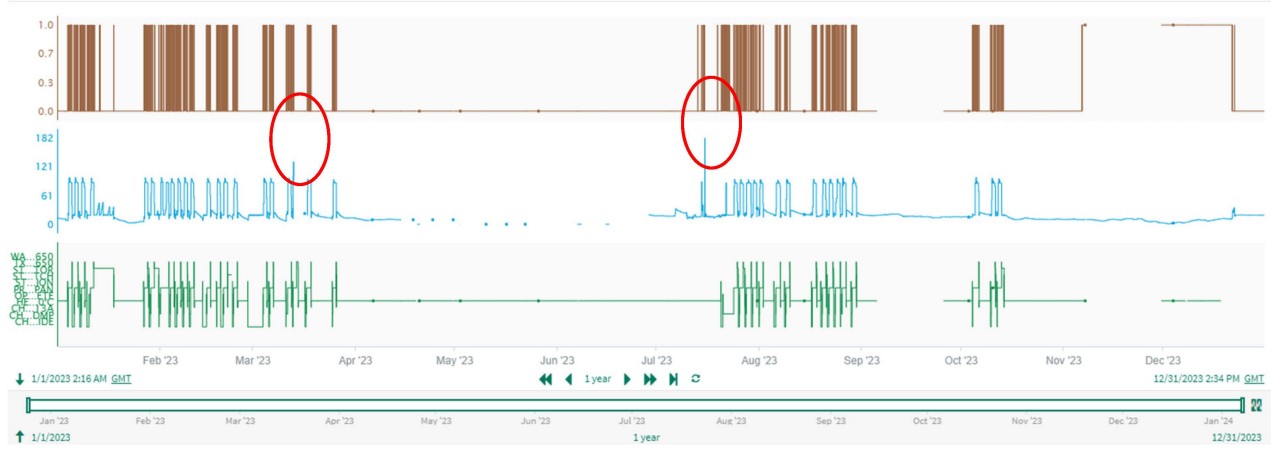

**Figure 10.** Instances where T-100 exceeds threshold.

Using the tools available in SeeQ R50 [10], zoom into the conditions created to identify those occasions and to show when the barrier was called upon, as shown by the red circle in Figure 11 and expanded in Figure 12 below:

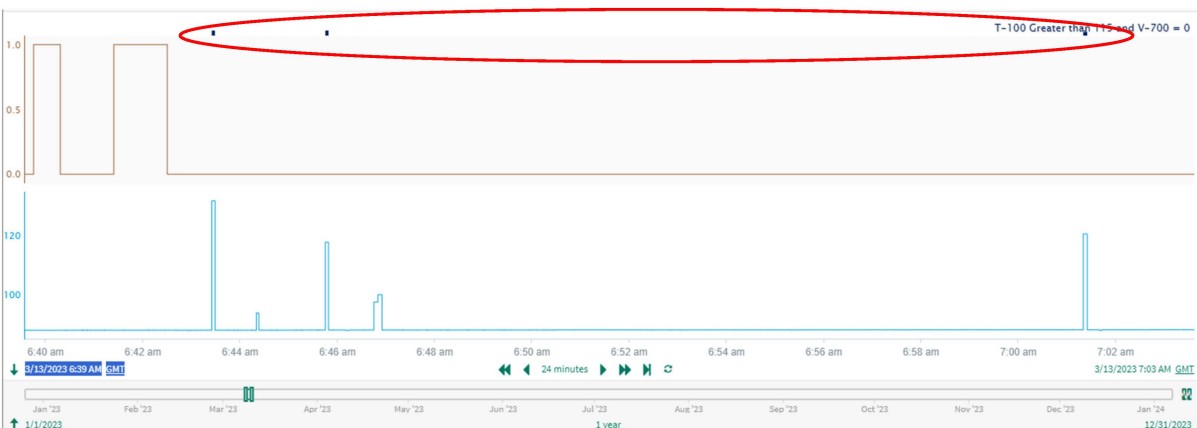

**Figure 11.** Conditions showing barrier action for the first occasion.

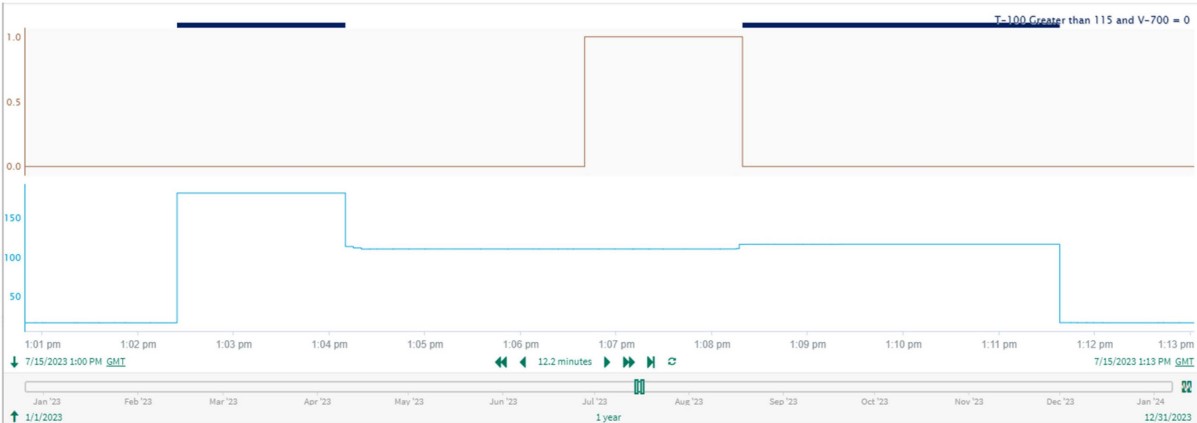

**Figure 12.** Conditions showing barrier action for the second occasion.

From the figures above, there were five different instances where the barrier was called upon and shown to be successful.

Data flows from the data lake to the software tool which is then manipulated to showcase the health of those barriers in an automated dashboard. The results of the dashboard can then be reviewed by the operation on a real- or near-real-time basis. The results of the dashboard could be updated with adjustable time periods. For the purposes of this paper, all the results from the models of the barriers use the same time period, the 2023 calendar year. All the results from the developed models were showcased in a live dashboard; the barriers that were called upon and were successful are presented in Figure 13 below:

| Name | Count | | | | | | | | | | |
|---|---|---|---|---|---|---|---|---|---|---|---|
| Count 1, No negative Rate, and Count 2 | 0 | | | | | | | | | | |

| Name | Count | LO | Name | Count | Name | Count | Name | Count | Name | Count | Name | Count |
|---|---|---|---|---|---|---|---|---|---|---|---|---|
| P<=1.5barg and V-200 and V-400 Closed | 2 | SS | Max Value of Stage 3 Reaction Exceeds 1.5d | 0 | T-100 Greater than 115 and V-700 = 0 | 5 | P>2.7 barg and V-500 = 0 | 1 | P>3.5 and Relief Valve Activated | - | P>3.85 and V-800 Open | - |

| Name | Count | RX | | | | | | | | | |
|---|---|---|---|---|---|---|---|---|---|---|---|
| Max Value per Stage 3 Reaction Greater than 8 kg | 0 | N | | | | | | | | | |

| Name | Count | RA | | | | | | | | | |
|---|---|---|---|---|---|---|---|---|---|---|---|
| Catalyst 1 valve closed during Catalyst 1 Charge Phase Waiting for Pushbutton | 0 | TE | | | | | | | | | |

**Figure 13.** BTDT dashboard in SeeQ organizer.

For clarity, the same results are reproduced in Table 4 below:

**Table 4.** Barrier results for 2023.

| Preventative Barrier | Count | | Mitigation Barrier | Count |
|---|---|---|---|---|
| 1 | 0 | | 1 | 0 |
| 2 | 2 | | 2 | 5 |
| 3 | 0 | Top Event | 3 | 1 |
| 4 | 0 | | 4 | 0 |
| | | | 5 | 0 |

The BowTie dashboard displays the results of the models when the barrier was called upon and successful.

## 5. Discussion and Conclusions

A BowTie is well suited as a safety digital twin. This paper demonstrates how organizations can develop safety management systems based on BowTies to monitor the risk space in real time throughout the organization. This yields an enormous span of control for safety management systems at a relatively low cost. Due to the data available, one is able to match appropriate data tags to the barriers and to structure the 'detect/decide/act' functionality with the data available.

This work shows that designing a BowTie from a data perspective changes the rules for making a BowTie and allows the determination of the health of the barrier system [32]. From a data perspective, it is more effective if different (right-hand side) mitigation barriers are aligned in a linear consequence line, where the order in which the barriers are triggered is the guiding principle: the lowest pressure-based barrier comes first and the higher pressures later, and temperature comes before pressure. Different endpoints (or 'consequences') are better represented as 'stopping points' articulated with the PEAR acronym for consequence types (people, environment, assets, and reputation). The order given by REAP is more sensible and is akin to a layer of a protection analysis structure, meaning that the barriers fail in order as per the design. To begin with, the barriers are in place to protect the organization's reputational interests; then, the barriers are in place to protect the environment and assets, leading to the final protections that are in place to prevent harm to personnel and bystanders. Also, the barriers that are in place are to protect the organization and their employees to ensure that they do not REAP the consequences. This is the key result of the work conducted in this project: a BowTie is a safety digital twin. Based on this statement, any organization can create a safety management digital twin by digitizing their BowTie for a process.

Organizations are now able to:

1. Monitor the health of barriers [14] with continuous monitoring of safety management systems, i.e., by counting the number of times the barrier was successful when called upon;
2. Be aware of the status of the process facility in real or near-real time whenever called upon;
3. Create more proactive measures and therefore create more leading PSPIs [30];
4. Allow management to make decisions which lead to positive actions;
5. Allow resource allocation to other tasks to add further value to the organization.

Thresholds can be included in the dashboard to then allow the organization to create red/amber/green rankings of the number of times a barrier has been called upon. These thresholds can then be reported to management, or they can indicate how often a barrier has been used. If a specific barrier is called upon on numerous occasions, the organizational personnel can then dive deeper into the root cause and attempt to remedy the process so that these conditions are not continuously being created.

As a corollary, the monitoring of 'detect/decide/act' functions within a barrier offers barrier health management without the actual testing of the barrier. This is derived from the description of preventative barrier 1 in Section 4.3.1. To assure ourselves that the barrier will operate when called upon, we want to monitor whether (a) the temperature sensor is operational (which we can check every second); (b) the PLC or DCS is online (which we can check every second and verify every internal diagnostic cycle); and (c) the valve is online (every second) and working in each operational cycle (a few times during the operation). This may reduce the frequency of the barrier functionality testing, which saves time and effort. Note that the degradation of the equipment associated with these functionalities may be tracked though remote conditioning, which gives an indication of the (degradation) reliability of the barrier as a system. This remains for further research. Controversially, it follows that it may actually be safer to depend on the continuous monitoring of operating process systems rather than installing independent safety layers that are never actually functioning. This also remains for further research.

Ultimately, digitization of the BowTie leading to the creation of a BTDT will lead to savings for the organization due to the reasons listed above. Due to the nature of the organization's produced product and the hazards and risks associated with what they produce, if that organization does not have the adequate safety management systems in place, they must REAP what they sow.

## 6. Recommendations, Limitations and Next Steps

From this work, further possibilities follow to use additional statistical methods for upper and lower control limits and to use regression analysis to further develop profiles and to use them to predict future trends. The inclusion of machine learning techniques can be used to determine whether the current values map onto those trends.

The BTDT for an organization has the ability to be inputted into the possibility of the creation of new PSPIs. These new PSPIs, due to the nature of the structure of the BTDT, would be leading PSPIs as the data stream in directly from the source. Additional focus can be put into the development of new leading PSPIs that have the potential to lead to the validation of the current barriers in place as well as to continuous improvement measures. For example, the organization can work on determining the understanding of the chemistry of the process, which could lead to the validation of acceptable threshold levels for the barriers in place.

From a single BTDT created from a single hazard and a top event, multiple BTDTs can be developed to represent that stage of the process. Going forward, all hazards and top events can also be created for that stage of the process and eventually the entire process. Multiple dashboards could be created to allow for a real-time view of the health of the process as the process proceeds. Essentially, the BTDT has the ability to provide a snapshot of the site. The SeeQ software application also possesses the ability to incorporate asset trees to link identical components which may be used on other parts of the process or other processes. This would allow the consolidation of a larger dataset, potentially for the computation of component reliability.

One of the limitations of this work is related to the current IT/OT landscape. Specific software tools are required to allow navigation of the IT/OT landscape for the BTDT development and the use of the current data storage tools. Another challenge associated with creating a BTDT is related to the possibility that the current data are not available in a readily available format such that information can be extracted or that tags can undergo any updates to the naming conventions, for example. Additional data sources, such as time-stamped databases would have to be created such that they were aligned with the data extraction method. With respect to new naming conventions, the entire model would have to be recreated with new tags that were spliced and connected with the previous tags. This study identified detect/decide/act relationships for the barriers listed, as if they were simple barriers; however, that may not be the case as not all barriers are simple barriers [33].

Detailed knowledge and access to process expertise is required to develop a concise BTDT with the identification of appropriate data tags and labels. With the use of resources that allow the development of the process BTDT, organizations will possess the ability to monitor the health of their processes in real or near-real time. Going forward, models have the possibility of being extended to compute component or barrier reliability and to update those values in real time using actual data from the process. Further validation of the BTDT requires future work on a larger scale across a process and potentially an entire site.

**Author Contributions:** Conceptualization, P.S. and C.v.G.; methodology, P.S.; software, P.S. and N.S.; validation, C.v.G. and N.S.; formal analysis, P.S.; investigation, P.S.; resources, P.S. and N.S.; data curation, P.S.; writing—original draft preparation, P.S.; writing—review and editing, P.S., C.v.G. and N.S.; visualization, P.S.; supervision, C.v.G. All authors have read and agreed to the published version of the manuscript.

**Funding:** This research received no external funding.

**Institutional Review Board Statement:** Not applicable.

**Informed Consent Statement:** Not applicable.

**Data Availability Statement:** Data is unavailable due to privacy and confidentiality restrictions.

**Conflicts of Interest:** Neil Sunderland is an employee of Syngenta Huddersfield Manufacturing Centre. All authors declare that this study received no funding from Syngenta Huddersfield Manufacturing Centre. Neil Sunderland had the following involvement with the study: Resources, Validation, Writing—review & editing.

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
