# Peer review of "The BowTie as a Digital Twin: How a BowTie Looks Different from a Data Perspective"

_safety, 2024_

Round 1

Reviewer 1 Report

Comments and Suggestions for Authors

This paper explores the integration of BowTie models into digital twin frameworks for enhanced safety management in chemical processing. It highlights how transitioning from traditional safety management approaches to digital twins can offer real-time monitoring and control, leveraging big data and advanced analytics to predict and mitigate risks.

Overall, I recommend publishing this paper after minor revision.

My detailed comments:

1. Ensure the introduction clearly outlines the research gap and how this study contributes to the body of knowledge. Expand on the theoretical underpinnings of digital twins and their application in safety management to provide a stronger foundation for readers unfamiliar with the concept.

2. Provide more detailed descriptions of the methodologies used for data analysis and the development of the BowTie digital twin. Specifics on software tools, data processing techniques, and model validation can enhance the replicability of the study.

3. Including detailed case studies or examples of implementing BowTie digital twins in actual chemical plants would greatly enrich the paper. These examples can illustrate the practical challenges and benefits encountered during implementation.

4. Offer clear directions for future research, particularly in areas where digital twins could be further optimized or applied to different sectors within chemical processing or beyond.

5. Enhancing the paper with more visuals, such as diagrams of the BowTie model transformations and screenshots of the digital twin interface, can aid in reader comprehension.

6. Ensure technical descriptions, especially those related to software and data analytics, are accessible to readers who may not have a deep background in these areas, possibly through the use of appendices or supplementary materials.

7. Two questions can be discussed further in the conclusion part:

Considering the dynamic nature of safety management in chemical processing, how does the BowTie digital twin model adapt to real-time data variations and unforeseen scenarios?

Can you elaborate on the model's scalability and its capability to integrate emerging technologies or methodologies for predictive analytics and risk management?

Reviewer 2 Report

Comments and Suggestions for Authors

Reviewer 3 Report

Comments and Suggestions for Authors

The paper presents an innovative approach to safety management systems, particularly in chemical processing plants. It explores the adaptation of the BowTie model into a digital twin to enhance real-time safety monitoring and risk management. The BowTie model's suitability as a digital twin for mapping out risk spaces and implementing real-life controls is emphasized, highlighting the need for specific rules and processes in designing and operating such a system effectively.

Here are some comments for brief responses:

1. The contribution and research work of the paper seem obscure. The paper does not look like a research work. The paper should explicitly state its novel contributions to the field. This includes distinguishing the proposed work from existing literature and clearly defining the advancements made in integrating the BowTie model with digital twin technology.

2. While the paper does a commendable job of integrating the BowTie model with digital twin technology, a deeper exploration of the theoretical underpinnings that guide the transition from a traditional safety model to a digital twin could enrich the narrative. Specifically, discussions on the limitations of traditional models in dynamic risk environments and how digital twins address these limitations would be beneficial.

3. The paper provides a detailed account of the process for creating a BowTie Digital Twin, yet it could benefit from a more structured comparison with other digital twin creation methodologies. This comparative analysis would offer readers a broader perspective on the advantages and potential drawbacks of the proposed method.

4. The paper effectively demonstrates how the BowTie Digital Twin can be used for process safety. However, a more critical analysis of the data generated by the digital twin, including its accuracy, reliability, and the potential for real-time intervention, would provide a more comprehensive understanding of its effectiveness.

5. The inclusion of case studies from actual industrial applications of the BowTie Digital Twin would significantly enhance the paper's practical relevance. These examples could showcase the challenges faced during implementation, strategies for overcoming these challenges, and the tangible benefits realized by organizations.

6. The paper concludes with recommendations and a path forward, but a more detailed discussion on the future of digital twins in safety management, including emerging technologies, potential new applications, and areas for further research, would be a valuable addition.

Comments on the Quality of English Language

Minor editing of English language required.

Round 2

Reviewer 1 Report

Comments and Suggestions for Authors

Accept the revised version.

Reviewer 2 Report

Comments and Suggestions for Authors

Most of the raised concerns have been adequately addressed, and I recommend proceeding with publication. However, the remaining doubt should be resolved. In Section 3.1, it is stated that the Bow-Tie visually represents all barriers, yet the types of safety barriers considered in the case study are limited. Further discussion is needed to explain whether barriers such as Human intervention, post-release physical protection, and others without sensors have the potential to be integrated into the framework of the Bow-Tie Digital Twin (BTDT).

Reviewer 3 Report

Comments and Suggestions for Authors

The quality has been improved.

Comments on the Quality of English Language

None.
